# Monitoring Physiological Performance over 4 Weeks Moderate Altitude Training in Elite Chinese Cross-Country Skiers: An Observational Study

**DOI:** 10.3390/ijerph20010266

**Published:** 2022-12-24

**Authors:** Yichao Yu, Ruolin Wang, Dongye Li, Yifan Lu

**Affiliations:** 1The School of Sports Medicine and Rehabilitation, Beijing Sports University, Beijing 100084, China; 2The Graduate School, Beijing Sport University, Beijing 100084, China; 3Melbourne School of Population and Global Health, The University of Melbourne, Carlton, VIC 3053, Australia; 4Key Laboratory of Sports and Physical Fitness of the Ministry of Education, Beijing Sport University, Beijing 100084, China

**Keywords:** cross-country skiing, physiological performance, moderate altitude training

## Abstract

The current observational study aimed to monitor the physiological performance over 4 weeks of living and training at a moderate altitude in elite Chinese cross-country skiers (8 males, mean age 20.83 ± 1.08 years). Lactate threshold, maximal oxygen uptake, blood, and body composition tests were performed at different time points to investigate the changes in physiological performance. The data were analysed by a one-way repeated measures ANOVA and a paired sample T-test between the test results. During the training camp, systematic load monitoring was carried out. Lactate threshold velocity, lactate threshold heart rate, and upper body muscle mass increased significantly (*p* < 0.01) after moderate altitude training. Maximum oxygen uptake was reduced compared to pre-tests (*p* < 0.05). Aerobic capacity parameters (maximal oxygen uptake, haemoglobin, red blood cell count) did not significantly increase after athletes returned to sea level (*p* > 0.05). These findings suggest that 4 weeks of moderate altitude training can significantly improve athletes’ lactate threshold and upper body muscle mass; no significant improvement in other aerobic capacity was seen. Exposure time, training load, and nutritional strategies should be thoroughly planned for optimal training of skiers at moderate altitudes.

## 1. Introduction

One of the most demanding endurance sports, cross-country skiing (XC-skiing), requires athletes to compete on courses between 1.5 km and 50 km long over various terrains. The different techniques of classical and skating styles require a high level of technical and physiological performance in athletes [1]. In sports training, physiological performance refers to the physiological adaptation of the athletes’ bodies for training or competition. Using physiological and biochemical indicators to monitor athletes’ performance has become an essential part of training in endurance sports [2].

Altitude training is widely used in endurance sports, which can induce different responses depending on the modality used [3,4]. It can generally be divided into three main types of training: living high training high (LH-TH), living high training low (LH-TL), and living low training high (LL-TH). Coaches and sports researchers use LH-TH to increase red blood cell (RBC) count, haemoglobin (Hb), maximal oxygen uptake (VO_2max_), and other performance measures at sea level [5]. However, athletes cannot maintain the same intensities as those at sea level. To avoid the limitations of LH-TH, LH-TL is used to improve athletes’ physiological performance while maintaining training intensity [6,7]. Some studies have found that LH-TL can enhance haematological and neuromuscular adaptations [8,9]. In LL-TH, athletes live at sea level, with hypoxic exposure lasting for several seconds to several hours during training and repeated over several days to weeks. LL-TH has been proven to improve athletes’ erythropoietin (EPO) and skeletal muscle mitochondrial density of hypoxia-inducible factor 1α (HIF-1α) [10,11]. The core aim of altitude training is to enhance the athletes’ physiological performance through hypoxic exposure in different ways. Meanwhile, unreasonable altitude training can lead to overreaching and even overtraining of athletes [12].

In recent years, many sports researchers have explored the effects of living and training at moderate altitudes (1500–3000 m), which is closer to the traditional LH-TH modality but lower in altitude [7,13,14]. From the point of view of combining exercise physiology and sports training, such an environment provides a certain level of hypoxic stimulation to the body while approaching the intensity of basic training, thus improving physiological performance [15,16]. Research has demonstrated that three weeks of training at 1800 m significantly increased haemoglobin levels in well-trained runners [17]. Czuba et al. [18] found that prolonged exposure to hypoxic conditions (simulated moderate altitude) would continue to promote the synthesis of erythropoietin, improving RBCs’ ability to carry more oxygen. Karlsson et al. reported that 17–21 days of training at 1800 m increased elite XC-skiers and biathletes’ lactate threshold velocity [19]. Four weeks of moderate altitude training (2200 m) increased the resting metabolic rate and haemoglobin in highly trained middle-distance runners [20]. The benefits of training at moderate altitudes include increased haemoglobin mass, body metabolic efficiency, and enhanced lactate threshold [21]. In the past two decades, most Olympic XC-skiing events have been held at 1500–1800 metres, and training at moderate altitudes has become common [22,23]. However, little systematic research has focused on changes in the physiological performance of XC-skiers at moderate altitudes.

This observational study aims to measure changes in physiological performance after four weeks of living and training at moderate altitude, meanwhile providing referenceable physiological evidence to help XC-skiing coaches and practitioners when using moderate altitude training. 

## 2. Materials and Methods

### 2.1. Subjects and Study Design

This research is an observational study of eight elite Chinese cross-country skiers over four weeks living and training at the Chinese national snow sports training base in BaShang, Chengde, Hebei Province (average sea level at 1510–1700 m, latitude at 44.5° N). A ‘polarised training’ plan based on a traditional Nordic XC-skiing training programme was used, and the daily training data were compiled by the scientists accompanying the team [23,24]. Throughout the design of the study, three tests were conducted (Figure 1): the pre-test (55 m above sea level) was completed five days before the moderate altitude training, the mid-test (1550 m above sea level) was performed on day 15 of the moderate altitude training, and the post-test (55 m above sea level) was conducted five days after the moderate altitude training.

The participants in this study included eight male athletes involved in preparation for the 2022 Winter Olympics in Beijing. The mean age was 20.83 ± 1.08 years; weight was 69.73 ± 5.12 kg; height was 179.63 ± 5.93 cm, and mean training years was 4.18 ± 1.92 years. All the athletes involved in the experiment were at the elite-level [25], were in excellent physical condition, and were uninjured. Prior to data collection, all athletes were given an informed permission form and a description of the experiment’s objective and potential dangers. Signing both agreements demonstrated a willingness to participate voluntarily. The Sports Science Experimental Ethics Committee of Beijing Sport University approved the research protocol (Grant No. 201906711).

### 2.2. Physiological Performance Test

Physiological performance tests in treadmill running were conducted using protocols developed by the Norwegian Top Sport Centre. First, an “intermittent” incremental test was used to obtain the lactate threshold. After a 5 min recovery, the athletes conducted an incremental test to determine VO_2max_ [26,27]. All athletes implemented a standardised warm-up process prior to the lactate threshold test under the supervision of a professional fitness coach. The warm-up routine consisted of 10 min of low-intensity jogging on a treadmill with an athlete rating of perceived exertion (RPE) of 2, followed by 10 push-ups and five squat jumps. After the athletes warmed up, the lactate threshold test was implemented on the treadmill (RL2500E, Rodby, Södertalje, Sweden) using the incremental load test method. The incline angle of the treadmill was set at 10.5% and maintained throughout the test, with the starting speed of the treadmill set at 7 km/h. Athletes ran at a constant speed for five minutes at each level of speed, with a 30 s rest interval at the end of the run. The treadmill’s speed was increased by 1 km/h between runs [26]. The heart rate level of each athlete was recorded in the last 30 s of each stage. The athlete’s blood lactate level was tested immediately after each level of the running platform test. Blood lactate concentration was measured immediately after exercise using an EKF benchtop blood lactate metre (Boisen, EKF Industrial Electronics, Magdeburg, Germany). Furthermore, the athlete’s RPE was recorded using a 0–10 scale. The lactate threshold was defined as a blood lactate level of 4 mmol/L^−1^ [28]; when the threshold exceeded 4 mmol/L^−1^, the test was stopped. Treadmill speed at 4 mmol/L^−1^ was calculated using linear interpolation [28].

Following the lactate threshold test, the athletes rested for 5 min before assessing their maximum oxygen uptake using a portable gas metabolism analyser (MetaMax 3B, Cortex, Leipzig, Germany) [26,27]. The treadmill incline angle for the VO_2max_ test was 10.5%, and the treadmill start speed was 1 km/h below the end speed of the lactate threshold test. The treadmill’s speed was increased by 1 km/h every minute from the beginning of the test until the participant was exhausted. Throughout the test, the athlete wore a ventilation mask to evaluate his oxygen uptake volume. The VO_2max_ was defined as the average of the two highest and consecutive 30 s measurements. A heart rate belt (H10, Polar, Finland) was used to monitor the athlete’s heart rate. Maximum HR was defined as the highest 5 s heart rate measurement during the VO_2max_ test. The blood lactate concentration was measured 1 min after completing the test, and the RPE values were recorded, with the RPE counted on a 0–10 scale. The athlete’s final treadmill speed, maximum oxygen uptake, and respiratory exchange ratio (RER) were recorded.

Blood tests were performed between 6:00 a.m. and 7:00 a.m. on each test day, with venous blood drawn by medical staff in the morning while the athlete was awake and fasting. Routine blood tests were analysed using a fully automated haematology analyser (BC-5180CRP Automatic Haematology Analyser, Myriad, Shenzhen, China). Blood urea (BUN) and creatine kinase (CK) were analysed using a fully automated biochemistry analyser (AU680 Automatic Biochemical Analyser, Beckman Coulter, Brea, CA, USA). All instruments were standardised using the original and matching reagents.

Body composition testing was conducted only on the morning of the pre-tests and post-tests. The fat, muscle, and bone mass of the athlete’s body and all body segments (upper body, trunk, and lower body) were measured using a dual-energy X-ray bone density analyser (Luna iDXA, General Electric Company, Schenectady, NY, USA) after the athlete had finished the venous blood draw.

### 2.3. Training Load Monitoring

This study’s four-week moderate altitude training programme involved six days of training per week, with Monday for rest and recovery. Low-intensity aerobic training was conducted five days before and after four weeks of moderate altitude training. All training loads, including volumes and intensities, were developed and implemented by the heart rate zones obtained from the athletes’ baseline physiological tests. Load monitoring used a 5-zone load intensity model (Table 1) designed by the Norwegian National Olympic Committee based on a combination of laboratory test results and actual training [24,29].

The training statistics format adhered to the training recording format suggested by the Norwegian National Olympic Committee, using exercise forms, training forms, and exercise intensity for load recording for all workouts (Figure 2) [24].

### 2.4. Statistical Analyses

Data statistics and analysis were performed using IBM SPSS 25.0 and Excel 2019 software. All data were presented as mean ± SD and were tested for normality using the Shapiro–Wilk test before processing. Differences in the lactate threshold test, maximum oxygen uptake test, and blood test between the three tests were assessed using repeated measures ANOVA. Mauchly’s Test of Sphericity was used, and in the multiple comparisons were used the Bonferroni post hoc test with significant changes at *p* < 0.05 and highly significant differences at *p* < 0.01. Differences in body composition between pre- and post-tests were using the paired sample T-test with significant changes at *p* < 0.05 and highly significant differences at *p* < 0.01.

## 3. Results

During four weeks of moderate altitude training, the eight elite Chinese male XC-skiers recorded a total training time of 67 h, with an average of 16.7 h per week. The average percentage of time spent in the i1-i5 intensity interval over four weeks was 63.6%, 28.5%, 5.8%, 1.4%, and 0.7%. The average LIT (i1–i2), MIT (i3), and HIT (i4–i5) intensities were 92.1%, 5.8%, and 2.1%, respectively. Endurance training throughout the cycle included roller skiing, skiing, running, and Nordic walking (Table 2).

According to Mauchly’s spherical hypothesis test, the variance–covariance matrix of all dependent variables was *p* < 0.01. The findings in Table 3 show the following. The athletes’ lactate threshold velocity, lactate threshold heart rate, VO_2max_, and maximal heart rate at VO_2max_ were statistically significant before and after the three tests (*p* < 0.01). The lactate threshold velocity was 0.24 m·s^−1^ higher in the post-test compared to the pre-test (95% CI: 0.046–0.436, *p* < 0.05) and 0.31 m·s^−1^ higher in the post-test compared to the mid-test (95% CI: 0.020–0.056, *p* < 0.05). The lactate threshold heart rate was 9.0 beats·min^−1^ higher in the post-test compared to the pre-test (95% CI: 5.079–12.921, *p* < 0.05) and 9.4 beats·min^−1^ higher in the post-test compared to the mid-test (95% CI: 0.020–0.056, *p* < 0.05). The VO_2max_ were 6.11 L·min^−1^ (95% CI: 2.293–9.932, *p* < 0.05) and 0.57 L·min^−1^ (95% CI: 0.213–0.917, *p* < 0.05) higher in the pre-test compared to the mid-test. The post-test result of VO_2max_ increased separately 3.50 mL·min^−1^·kg^−1^ (95% CI: 0.545–6.455, *p* < 0.05) and 0.39 L·min^−1^ (95% CI: 0.108–0.674, *p* < 0.05) compared to the mid-test. The maximum heart rate of the athletes tested was 9.0 beats·min^−1^ higher in the post-test than in the mid-test (95% CI: 2.763–9.23, *p* < 0.01).

The heart rate–velocity and lactate–velocity curves plotted from the lactate threshold test data are shown in Figure 3. After training at moderate altitude (post-test), the lactate threshold heart rate was significantly higher than pre-test and mid-test at all speed levels (*p* < 0.05).

As demonstrated in Table 4, there were significant changes (*p* < 0.05 or *p* < 0.01) in RBC, BUN, and %MXD in the three tests. The RBC increased by 0.29 × 10^6^·μL^−1^ in the pre-test compared to the mid-test (95% CI: 0.041–0.534, *p* < 0.05). The BUN increased 1.16 mmol·L^−1^ after two weeks of exposure to the moderate altitude (mid-test) compared to the pre-test (95% CI: 0.195–2.130, *p* < 0.05). The % MXD increased separately by 3.26% and 3.36% in the pre-test (95% CI: 1.246–5.279, *p* < 0.01) and mid-test (95% CI: 1.720–5.005, *p* < 0.01) compared to the post-test. For HCT and WBC, the F-test achieved statistical significance. There was no significant difference between the measurements. This could happen if the sample size is small, leading to unstable statistical results

A paired sample T-test of body composition data during four weeks at moderate altitude (Table 5) showed that the percentage of upper-body muscle mass increased from 11.74% to 12.03% after altitude training, with a significant increase in upper-body muscle mass before and after training (95% CI: 0.093–0.307, *p* < 0.01).

## 4. Discussion

The primary contribution of our study is to provide relevant and new data regarding the effects of four weeks of moderate altitude training on the physiological performance of elite Chinese XC-skiers. This study indicates that four weeks of living and training at a moderate altitude leads to elevated lactate threshold and upper-body muscle mass.

In the case of lactic acid build-up, the athletes’ cells’ removal of lactic acid somewhat reflects the athletes’ aerobic and lactate metabolism capacity [30]. This study found that lactate threshold velocity and heart rate were significantly higher after four weeks of moderate altitude training, reflecting an increase in the athletes’ lactate threshold and aerobic capacity. The lactate–velocity curve shows that the blood lactate concentration after four weeks of moderate altitude training was consistently lower than the first two tests at the velocity level prior to reaching the lactate threshold. This improvement is consistent with Ingjer et al. [31], who discovered that excellent XC-skiers had significantly reduced lactate levels in the submaximal test after three weeks of moderate altitude training at 1900 m. In addition, a study of a group of British national team distance runners showed a 12% increase in lactate threshold velocity after four weeks of 1500–2000 m endurance training [32].

High biochemical blood indicators are the primary reason for applying altitude training to increase lactate metabolism [33]. However, no similar situation occurred in this study. Unlike the traditional LH-TH plan (over 3000 m), 1550 m may be too low to increase biochemical blood indicators, such as Hb [19]. We speculate that the increase in lactate metabolism capacity has resulted from non-haematological hypoxia-induced changes. In LH-TL, elite athletes living at high altitudes of 2000–3000 m while simultaneously training below 1500 m can enhance muscle buffering capacity [34]. In this study, the living altitude of athletes was much lower, which is more conducive to recovery and may further improve muscle buffering capacity and lactate metabolism. Meanwhile, with the continuous promotion of training, the further improvement of the athletes’ economy of action may also have some impacts. Based on the current experimental design, we could not determine whether the effect was due to hypoxic exposure or a training plan.

Maximal oxygen uptake is one of the most important indicators of XC-skiers [1,35,36]. We found that the maximum oxygen uptake test on the plateaus was significantly lower than on the plains. This is common in altitude training; as the partial pressure of oxygen decreases, oxygen from arterial blood is difficult to deliver to tissue cells, with negative effects on muscle metabolism and contraction, leading to increased peripheral fatigue [12]. Contrary to the traditional LH-TH plan, maximal oxygen uptake levels do not change significantly after returning to sea level. Ingjer et al. [31] and Chapman et al. [37] also found the same results in elite endurance athletes when they were training at 2000–3000 m. Compared with the altitude used in the traditional altitude training modality, this may be because a moderate altitude environment does not cause a strong enough stimulus to the athletes’ bodies [19]. In addition, insufficient training intensity may lead to this phenomenon. According to the training load monitoring results, the lack of HIT training may have contributed to the lack of cardiorespiratory and physiological stimulation. In addition to no changes in physiological indicators, training at 1550 m may not suffer any change in lung diffusing, which was proved in 1800 m swimming training [38]. A detailed examination applied in a meta-analysis about altitude training reveals a lag in the change in maximal oxygen uptake after altitude training. The longer the time after returning to sea level, the more pronounced the increase in maximal oxygen uptake due to altitude training [3]. Whether similar mechanisms exist at this altitude could be further verified by future experiments.

RBCs are the body’s carriers of oxygen and play an important role in endurance training. This study found that after two weeks of moderate altitude training stimulation, athletes’ RBC counts and haemoglobin levels tended to decrease compared to the pre-test. After returning to sea level, neither increased relative to pre-training levels. The same result was reported in a study with XC-skiers under long-term moderate altitude training (1500–1800 m) [39]. While no EPO could be measured in the present study, the study of altitude training indicated that higher altitude had a more positive effect on RBC production in athletes [9,40]. Meanwhile, several prior studies have reported a correlation between maximal oxygen uptake values and haemoglobin concentrations in athletes [31]. Maximal oxygen uptake and haemoglobin change trends in this case study are also relevant. Contrary to previous research in other sports, our results indicate that training at 1550 m may not stimulate red cell production with concurrent amelioration of aerobic performance like VO_2max_. The insufficient hypoxia exposure and some hypoxia-induced disturbances in physiological function will cause it [40].

At the same time, attention should be paid to the exposure time to hypoxia and the environment; the terrain, wind speed and load could have altered this outcome. The training camp environment is windy all year round, which may lead to the loss of athletes’ body fluids, resulting in negative effects. Typically, the duration of moderate altitude training is generally 7 to 21 days [23], and 28 days of hypoxic exposure may have a negative impact. Moreover, we did not control plasma and blood volume, which may have affected the results. BUN is more sensitive to changes in training volume, as the higher the training load, the greater the rise in BUN and the slower the recovery rate early the following morning [12]. In the mid-test of this study, BUN was significantly higher than in the pre-test, which is also consistent with the accumulation of fatigue in athletes as the load increases. After moderate altitude training, the percentage of monocytes in the athletes in this study was much lower than before, and throughout training, the athletes’ bodies likely showed signs of infection-like conditions at the beginning, and the accumulated load of altitude training contributed to this phenomenon [32].

There is evidence that athletes exposed to hypoxic environments for training may experience significant changes in body composition [41]. MacDougall J et al. [42] found that long-term exposure to chronic hypoxia at high altitudes (over 5000 m) can disrupt the body’s protein synthesis, which in turn can result in lower body weight, skeletal muscle mass, and fat mass. In contrast, we did not find significant body mass, bone, or muscle loss in this case study, which is likely to be related to insufficient exposure time. The upper-body muscle mass change showed an upward trend after four weeks of moderate altitude training. It is worth noting that during this period, the Chinese XC-ski team coaching team specifically strengthened the athletes’ strength base. The training load intervention created a certain intervention mechanism on the body, resulting in the athletes’ body muscle mass being well maintained, which has been proven in elite winter sports athletes [43,44]. Another possible explanation is that specialist nutrition supplies from the national team may have led to this result. It is dangerous for endurance athletes to lose muscle and body weight during altitude training. Nevertheless, a solid nutritional approach prevents significant changes in body composition [45,46]. The study of Kayser et al. [47] has demonstrated that it is possible to maintain body composition at altitudes below 5000 metres if people intake sufficient energy. These results suggest that the athletes could maintain energy balance throughout the moderate altitude camp. Both training load and nutritional strategy may have influenced the results of this study, which can be verified in the future through well-controlled studies.

Some limitations of this study should be considered. Firstly, Due to the limitations of the design, this study lacked a control group, and the observations in this study were relatively small. Nevertheless, this issue is a standard limitation of observational studies in real-world competitive sports settings. Secondly, there was a high degree of uncertainty in training during the field follow-up observations, and we needed to fully account for these confounding factors’ effects. Thirdly, due to realistic conditions, we have only discussed the physiological performance of athletes. Competitive sports are a results-driven business, and more data on sports performance are needed. Regarding further research, it would be interesting to replicate the present study with more athletes, different genders, different exposure times at moderate altitudes, or even different kinds of altitude training modalities with a control group.

## 5. Conclusions

Four weeks of training at a moderate altitude positively affected the athletes’ lactate threshold and upper-body muscle mass. Maximum oxygen uptake was reduced in athletes tested at high altitudes compared to sea level due to being in a hypoxic environment. The aerobic capacity indicators (maximal oxygen uptake, haemoglobin, and RBC count) did not improve significantly after the athletes returned to sea level compared to pre-moderate altitude training, which may be related to altitudes that were too low, the duration of exposure to the hypoxic environment, and the training load design. When imposing moderate altitude training on athletes, exposure time, training load characteristics, and nutritional strategies must be meticulously designed to optimise training outcomes.

## Figures and Tables

**Figure 1 ijerph-20-00266-f001:**
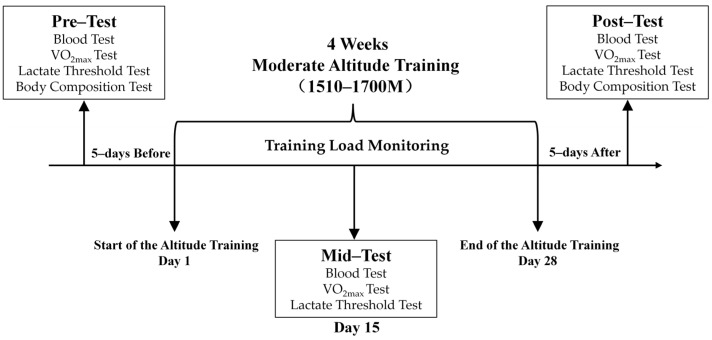
Study Design of the Research.

**Figure 2 ijerph-20-00266-f002:**
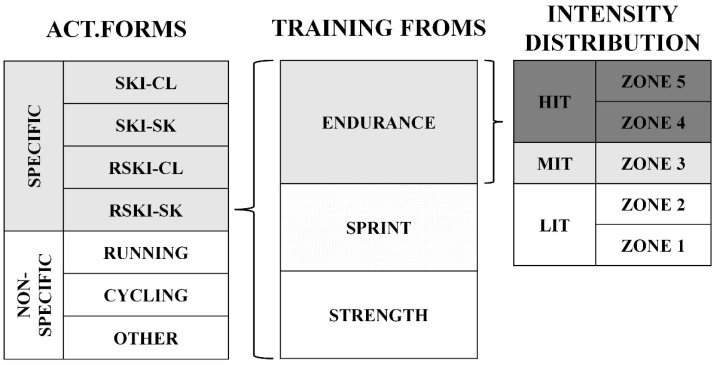
Training distribution methods. abbreviations: ACT. FORMS = activity forms.

**Figure 3 ijerph-20-00266-f003:**
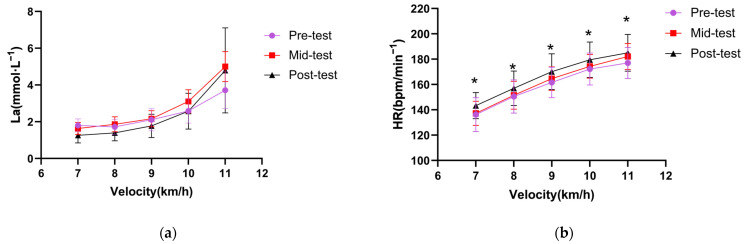
Eight elite Chinese cross-country skiers’ lactate-velocity curve (**a**) and heart rate-curve (**b**) from lactate threshold test. * indicates a significant difference compared to pre-test (* *p* < 0.05). Abbreviations: La = lactate, HR = heart rate.

**Table 1 ijerph-20-00266-t001:** The 5-zone model was used in the current study.

Intensity Zone		Blood Lactate (mmol/L)	Heart Rate (% Max)	RPE
5	HIT	6.0–10.0	92–97	8–10
4		4.0–6.0	87–92	7–9
3	MIT	2.5–4.0	82–87	4–7
2	LIT	1.5–2.5	72–82	3–5
1		0.8–1.5	55–72	≤4

Abbreviations: LIT: low-intensity training; MIT, moderate-intensity training, HIT, high-intensity training.

**Table 2 ijerph-20-00266-t002:** Training Characteristics of 4 weeks Moderate Altitude Training.

Weekly Training Patterns	Week1	Week2	Week3	Week4
**Total Training**				
Total Training Time (h·wk^−1^)	13.0 ± 0.6	17.4 ± 1.2	16.9 ± 2.7	19.7 ± 1.4
Training Sessions·wk^−1^	10.4 ± 1.1	12 ± 2.3	10.5 ± 2.4	12.6 ± 1.4
Endurance Distance (km·wk^−1^)	155.4 ± 19.5	205.3 ± 35.1	208.9 ± 38.3	237.6 ± 17.6
**Training forms**				
Endurance Training Time (h·wk^−1^)	11.8 ± 1.0	14.3 ± 1.2	14.2 ± 1.6	16.9 ± 0.7
Strength Training Time (h·wk^−1^)	1.1 ± 0.9	3 ± 0.5	2.5 ± 1.3	2.7 ± 1.1
Sprint Training Time (h·wk^−1^)	0.1 ± 0.1	0.2 ± 0.1	0.2 ± 0.1	0.1 ± 0.1
**Endurance Intensity distribution**				
Zone 1 (h·wk^−1^)	7.5 ± 1.6	9.2 ± 3.4	9.9 ± 2.4	9.8 ± 3.2
Zone 2 (h·wk^−1^)	3.3 ± 1.5	3.9 ± 1.8	3.7 ± 2.5	5.4 ± 2.6
Zone 3 (h·wk^−1^)	0.6 ± 0.4	0.8 ± 1	0.5 ± 0.4	1.4 ± 0.9
Zone 4 (h·wk^−1^)	0.2 ± 0.2	0.3 ± 0.3	0.1 ± 0.1	0.2 ± 0.2
Zone 5 (h·wk^−1^)	0.2 ± 0.1	0.1 ± 0.1	0 ± 0.1	0.1 ± 0.1

**Table 3 ijerph-20-00266-t003:** Changes in Lactate Threshold and Maximum Oxygen Uptake after 4 weeks Moderate Altitude Training.

	Pre-Test	Mid-Test	Post-Test	Time Effect
Lactate threshold velocity(m·s^−1^)	3.02 ± 0.18	2.96 ± 0.14	3.27 ± 0.24 *^$^	F(2,14) = 10.40, *p* = 0.002
Lactate threshold HR(beats·min^−1^)	179.5 ± 9.8	179.1 ± 10.3	188.5 ± 9.2 **^$$^	F(2,14) = 26.46, *p* < 0.001
VO_2max_ (mL·min^−1^·kg^−1^)	73.74 ± 3.63	67.63 ± 2.13 **	71.12 ± 3.14 ^$^	F(2,14) = 15.48, *p* < 0.001
VO_2max_ (L·min^−1^)	4.82 ± 0.36	4.25 ± 0.36 **	4.64 ± 0.43 ^$^	F(2,14) = 19.70, *p* < 0.001
RER	1.23 ± 0.09	1.23 ± 0.05	1.11 ± 0.03 **^$^	F(2,14) = 14.53, *p* < 0.001
Maximum HR(beats·min^−1^)	197.4 ± 11.6	192.9 ± 8.6	198.9 ± 9.1 ^$$^	F(2,14) = 9.41, *p* = 0.003
Maximum Lactate(mmol·L^−1^)	13.31 ± 1.42	11.69 ± 2.34	11.33 ± 2.79	F(2,14) = 1.84, *p* = 0.196

* indicates a significant difference compared to pre-test (* *p* < 0.05, ** *p* < 0.01). ^$^ indicates a significant difference compared to mid-test (^$^
*p* < 0.05, ^$$^
*p* < 0.01). Abbreviations: HR = heart rate, VO_2max_ = maximal oxygen uptake, RER = respiratory exchange ratio.

**Table 4 ijerph-20-00266-t004:** Changes in Blood Indicators after 4-weeks Moderate Altitude Training.

	Pre-Test	Mid-Test	Post-Test	Time Effect
HCT	0.48 ± 0.02	0.46 ± 0.02	0.46 ± 0.02	F(2,14) = 6.85, *p* = 0.008
HGB/(g·L^−1^)	160.00 ± 10.90	156.00 ± 7.25	160.88 ± 10.48	F(2,14) = 2.71, *p* = 0.101
RBC/(×10^6^·μL^−1^)	5.40 ± 0.32	5.11 ± 0.29 *	5.23 ± 0.27	F(2,14) = 8.70, *p* = 0.004
CK/(U·L^−1^)	207.63 ± 122.47	233.38 ± 101.77	181.38 ± 80.15	F(2,14) = 1.46, *p* = 0.265
BUN/(mmol·L^−1^)	6.65 ± 1.10	7.81 ± 0.74 *	6.71 ± 0.96	F(2,14) = 5.96, *p* = 0.013
%LYM	38.6 ± 6.3	39.3 ± 8.2	40.6 ± 7.8	F(2,14) = 0.39, *p* = 0.683
%MXD	7.8 ± 2.7	7.9 ± 1.8	4.6 ± 1.4 **^$$^	F(2,14) = 16.26, *p <* 0.001
%NEUT	53.6 ± 6.3	50.8 ± 9.0	53.1 ± 9.3	F(2,14) = 18.41, *p* = 0.518
WBC/(×10^3^·μL^−1^)	5.84 ± 1.30	4.93 ± 0.84	5.35 ± 0.63	F(2,14) = 4.86, *p* = 0.025

* indicates a significant difference compared to pre-test (* *p <* 0.05, ** *p <* 0.01). ^$^ indicates a significant difference compared to mid-test (^$$^
*p <* 0.01). Abbreviations: HCT = hematocrit, HGB = hemoglobin, RBC= red blood cell, CK = creatine kinase, BUN = blood urea nitrogen, %LYM = percentage of lymphocyte, %MXD = percentage of mononucleosi, %NEUT = percentage of neutrophil, WBC = white blood cell.

**Table 5 ijerph-20-00266-t005:** Changes in Body Composition after 4-weeks Moderate Altitude Training.

	Pre-Test	Post-Test	*p* Values
**Muscle mass (kg)**			
Whole body	55.66 ± 4.01	56.01 ± 4.02	*p* = 0.071
Upper body	6.54 ± 0.42	6.74 ± 0.36 ^##^	*p* = 0.004
Trunk	27.20 ± 2.18	27.47 ± 2.00	*p* = 0.485
Lower body	18.60 ± 1.82	18.57 ± 1.68	*p* = 0.924
**Fat mass (kg)**			
Whole body	7.81 ± 1.51	7.91 ± 1.14	*p* = 0.193
Upper body	0.91 ± 0.16	0.97 ± 0.18	*p* = 0.172
Trunk	3.13 ± 0.84	3.06 ± 0.60	*p* = 0.649
Lower body	2.91 ± 0.56	3.03 ± 0.47	*p* = 0.139
**Bone mass (kg)**			
Whole body	2.83 ± 0.16	2.84 ± 0.13	*p* = 0.356
Upper body	0.40 ± 0.01	0.41 ± 0.04	*p* = 0.172
Trunk	0.84 ± 0.05	0.81 ± 0.04	*p* = 0.356
Lower body	1.16 ± 0.08	1.16 ± 0.08	*p >* 0.05

^#^ indicates a significant difference compared to pre-test (^##^
*p <* 0.01).

## Data Availability

Not applicable.

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
