# Peer review of "Monitoring Physiological Performance over 4 Weeks Moderate Altitude Training in Elite Chinese Cross-Country Skiers: An Observational Study"

_ijerph, 2022, doi:10.3390/ijerph20010266_

Round 1
Reviewer 1 Report (New Reviewer)
Please see attached

Author Response
Please see the attachment.
We have now added training information to the paper. But based on this, we would like to apply to change the manuscript format form Case Report to Article. Because as a case report, there is not enough space for a description.

Reviewer 2 Report (New Reviewer)
This case report provide helpful results regarding the physiological changes after a moderate altitude training. Some minor comments have to be addressed before publication:
- Introduction: have the altitude training any harmful effects on athletes? If so, please add this information in this section (lines 38-52)
- Methods: very well described. Congratulations
- Results: Figure 1 seems to be a little blurred. Please provide it in a high quality in order to analyse the results properly.
- Discussion: Considering that the main point of this study is the novelity of the sample (elite XC-ski athletes), authors should discuss the specific advantages of the altitude training and the physiological changes in this population.
- References: please review the list of references according to the journal guidelines.
Round 2
Reviewer 1 Report (New Reviewer)
Concerning the submitted manuscript, the authors did extensive review and additions to the manuscript Monitoring Physiological Performance over 4 weeks Moderate Altitude Training in Elite Chinese Cross-country Skiers: An Observational Study.
I am impressed with the amount of revisions and additions the authors have made to their manuscript. It is indeed a more complete story of the interesting altitude training and testing of elite level cross country skiers prior to Olympics. This is a noteworthy observational study, especially of elite level athletes, and worthy of publication. I only have a small number of suggestions below. There may still be some unit, table, and/or figure formatting issues specific for the Int. J. Environ. Res. & Public Health.
1. Purpose statement is not complete sentence …. “to provide evidence support “?
2. Appreciate Figure 1. A nice Time line was added to the study.
3. Line 603. “Athletes RAN …”
4. Figure 2. What is the abbreviation ACT.? Can you put this abbreviation into the figure legend? I really like this training volume table and it certainly adds to the study.
5. Table 2. Second row. What are Training Session (wk-1)? Is this a number? So, it would be (#/week)? So, from 10 to about 12 sessions in a week? Maybe change units to (#/week). I really like this training volume table and it certainly adds to the study.
Author Response
Please see the attachment.
According to your review report and suggestions during peer review, we would like to apply to change the manuscript type form "Case Report" to "Article". According to academic standards, as a case report there is not enough space for a description of our topic.

This manuscript is a resubmission of an earlier submission. The following is a list of the peer review reports and author responses from that submission.
Round 1
Reviewer 1 Report
Thank you for the opportunity to review again this manuscript. As a case report this article could be considered, however manuscript quality is too low to reach standard of publication. Details are in my following comments:
Overall, the manuscript NEEDS a professional English proof reading. Grammar is uncorrected and there is a lot of typing mistake (e.g.: missing space between word, missing word …) making the manuscript difficult to read and understand.
Moreover, many sentences are useless and inappropriate in a scientific paper: Author used terms like “physiological capacities” “specific athletic abilities” or “a certain mechanism” that refer to nothing. Be specific throughout and detail the mechanism, abilities, capacities you are referring to.
Introduction:
L56: “But so far, the limited research findings to date hardly prove that live and train at moderate 57 altitude is the best way to induce beneficial responses.”
This statement reflects the main weakness of the introduction. Authors should have a better comprehension of altitude training design and efficiency based not only on XC-skiing study. Their conclusions are wrong and unsupported by the current literature.
Authors should clearly state the altitude used in the study which are quoted in the introduction and discussion.
Method:
The location of the training camp should be specified. Indeed, latitude is also a factor that change barometric pressure.
Blood lactate test is not properly described (miss the speed increment between each run).
Authors use present and past alternatively in the methods, be consistent.
Results:
Results section is not understandable, I cannot know when p value refers to main effect or paired comparison.
Authors report “trend” without any p value which is not acceptable. Please report only significant result.
Table 1: The hashtag symbol is not defined.
Table 3: missing p value for the last line.
Discussion:
Authors report that the altitude exposure was simulated, if it’s the case the method should be adapted.
Authors based some part of their discussion on non-significant results. These parts may be deleted. Moreover, authors report results from other paper but did not use it to discuss their own findings.
Author should discuss the role of altitude severity and training modality (LH-TL, LL-TH…) in training adaptation.
The lack of a control group should be mentioned as the primary limitation.
Reviewer 2 Report
I believe that is a well-designed and well-conducted study. I believe it can be accepted for publication after minor corrections:
Materials and mehods
line 73 remove parentheses from age, wight, height, etc.
Throughout the text remove the initial of the first name and leave only the last name followed by et al., unless two last names are the same.
Reviewer 3 Report
I congratulate the authors for making the suggested changes.